# Avialan-like brain morphology in *Sinovenator* (Troodontidae, Theropoda)

Congyu Yu [1,2,3], Akinobu Watanabe[3,4,5], Zichuan Qin[6], J. Logan King[6,7], Lawrence M. Witmer[8], Qingyu Ma[9] & Xing Xu [7,10,11]✉

Many modifications to the skull and brain anatomy occurred along the lineage encompassing non-avialan theropod dinosaurs and modern birds. Anatomical changes to the endocranium include an enlarged endocranial cavity, relatively larger optic lobes that imply elevated visual acuity, and proportionately smaller olfactory bulbs that suggest reduced olfactory capacity. Here, we use micro-computed tomographic (μCT) imaging to reconstruct the endocranium and its neuroanatomical features from an exceptionally well-preserved skull of *Sinovenator changii* (Troodontidae, Theropoda). While its overall morphology resembles the typical endocranium of other troodontids, *Sinovenator* also exhibits unique endocranial features that are similar to other paravian taxa and non-maniraptoran theropods. Landmark-based geometric morphometric analysis on endocranial shape of non-avialan and avialan dinosaurs points to the overall brain morphology of *Sinovenator* most closely resembling that of *Archaeopteryx*, thus indicating acquisition of avialan-grade brain morphology in troodontids and wide existence of such architecture in Maniraptora.

[1] State Key Laboratory of Oil and Gas Reservoir Geology and Exploitation & Institute of Sedimentary Geology, Chengdu University of Technology, Chengdu 610059, China. [2] Key Laboratory of Deep-time Geography and Environment Reconstruction and Applications of Ministry of Natural Resources, Chengdu University of Technology, Chengdu 610059, China. [3] Division of Paleontology, American Museum of Natural History, New York, NY 10024, USA. [4] Department of Anatomy, New York Institute of Technology College of Osteopathic Medicine, Old Westbury, NY 11568, USA. [5] Department of Life Sciences, Natural History Museum, London SW7 5BD, UK. [6] Palaeontology Research Group, School of Earth Sciences, University of Bristol, Bristol BS8 1RJ, UK. [7] Key Laboratory of Vertebrate Evolution and Human Origins of Chinese Academy of Sciences, Institute of Vertebrate Paleontology and Paleoanthropology, Chinese Academy of Sciences, Beijing 100044, China. [8] Department of Biomedical Sciences, Heritage College of Osteopathic Medicine, Ohio Center for Ecological and Evolutionary Studies, Ohio University, Athens, OH 45701, USA. [9] Chongqing Laboratory of Geological Heritage Protection and Research, No. 208 Hydrogeological and Engineering Geological Team, Chongqing Bureau of Geology and Minerals Exploration, Chongqing 401121, China. [10] Centre for Vertebrate Evolutionary Biology, Yunnan University, Kunming 650091, China. [11] Paleontological Museum of Liaoning, Shenyang Normal University, Liaoning Province, 253 North Huanghe Street, Shenyang 110034, China. ✉email: xu.xing@ivpp.ac.cn

During the last three decades, substantial advances have been made in studying the origin and early evolution of birds. Evidence from paleontology, embryology, and molecular evolution all support that birds arose from a group of feathered non-avialan dinosaurs similar to Troodontidae and Dromaeosauridae[1–3]. The fossil records suggest a gradual acquisition of avian characteristics[4], and there have also been well-recorded examples of mosaic evolution in birds and their closest relatives, such as decoupled osteological features from forelimbs, pelvis, crania, and hindlimbs in troodontids[5], shoulder girdle ossification[6], and cranial kinesis[7]. What have been historically considered 'avian' traits have also independently evolved in multiple non-avialan theropod lineages, for example, the pygostyle in Oviraptorosauria[8], loss of forelimb digits[9], and powered flight among paravian dinosaurs[10].

A major anatomical transformation along the transition from non-avialan to avialan dinosaurs also occurs in their brain morphology. In paleoneurology, the internal molds of the braincase, also known as cranial endocasts ("endocasts" hereafter), are typically used as proxies for brain size and shape of extinct vertebrates[11–15]. More recently, computed tomography (CT) imaging has greatly facilitated paleoneurological research by allowing endocranial reconstructions through non-invasive means[16]. Rogers[17] showed the anatomy of an exceptionally well-preserved cranial endocast of *Allosaurus* on the basis of spiral CT scanning and demonstrated that this genus had a crocodilian-like neuroanatomy. Knoll et al.[18] made the first digital reconstruction of a large theropod dinosaur braincase. Early comparative work includes Larsson et al.[19], who studied the forebrain of *Carcharodontosaurus* and *Tyrannosaurus* based on CT and surface laser scans, and suggested a proportionally small brain in these macropredators compared to *Archaeopteryx* and extant birds. Witmer et al.[20] reconstructed the endocasts of two pterosaurs, *Rhamphorhynchus* and *Anhanguera*, and compared them with other archosaurs, showing convergent and distinctive features between birds and pterosaurs. Balanoff et al.[21] provided a large-scale quantitative comparative study on theropod dinosaur endocasts by sampling specimens from Oviraptorosauria, Deinonychosauria, and multiple lineages of extant birds. The results showed that brains that exhibit aspects of avialan brains evolved multiple times among maniraptoran theropods and that the brain of *Archaeopteryx* was not as uniquely specialized as previously thought, suggesting an earlier occurrence of avialan-like brain architecture among non-avialan dinosaurs. Currently, approximately 150 non-avialan dinosaur species have had their brain structures reported, in which many were based on high-resolution 3D imaging and cover all major lineages[22]. But among the reported endocrania, only a few are from paravian taxa, particularly those early branching species, thus obscuring the link between typical non-avialan theropod brains and those of birds.

*Sinovenator* is a troodontid dinosaur reported by Xu et al.[23] based on two partial specimens, including a disarticulated partial skull from the Early Cretaceous strata in Liaoning, northeastern China. It was originally recovered as the most basally divergent troodontid with subsequent phylogenetic studies also supporting *Sinovenator* as a basal troodontid[5,10,23–25]. A short description of the skull (including the braincase) and postcranial elements was published by Xu et al.[23]. The original reconstruction – which was based on the partial skull specimen IVPP V12615 – illustrated its braincase from the right lateral view and suggested a typical troodontid morphology based on the absence of a basisphenoid recess[23].

Yin et al.[26] presented a more detailed description of *Sinovenator*'s cranial morphology based on another partial skull from specimen PMOL-AD00102. This specimen preserves most of the cranial elements that are posterior to the antorbital fenestra and

six articulated cervical vertebrae. Digital reconstruction from CT scans shows that the posterior region of the braincase preserves most of its elements, and is mildly distorted transversely and dorsoventrally[26]. Yin et al.[26] suggested that the morphology of the *Sinovenator* braincase was intermediate between late-diverging troodontids (such as *Almas* and *Sinusonasus*) and non-troodontid paravians, primarily due to *Sinovenator* bearing a small subotic recess (a trait that was not consistent with Xu et al.[23] on specimen IVPP V12615) and the presence of an otosphenoidal crest. Besides the detailed description that predominantly focused on the osteological morphology by Yin et al.[26], the authors did not reconstruct the cranial endocast nor did they quantitatively compare either specimen to other theropod specimens.

In this study, we report on a new three-dimensionally preserved skull of *Sinovenator changii* (Troodontidae, Theropoda) and focus on its endocranial morphology. Detailed description on different parts of the cranial endocast and endosseous labyrinth is made with other closely related taxa. A landmark-based geometric morphometric (GM) analysis is conducted to quantitatively compare the endocranial shape of *Sinovenator* to that of other non-avialan theropod dinosaurs ($n = 11$), *Archaeopteryx*, and extant birds ($n = 38$), with the American alligator (*Alligator mississippiensis*) as an outgroup. Given its phylogenetic position as a basal troodontid, the brain structure of *Sinovenator* will help elucidate the tempo and mode of brain evolution at a critical point when theropod dinosaurs began to acquire features of modern avian brains[21,27,28].

## Results

IVPP V20378 is a nearly complete skeleton of *Sinovenator changii*, preserved in a sleeping posture (Fig. 1) resembling another troodontid, *Mei long*[29]. Although the skull is isolated from the skeleton, most of the postcranial elements are articulated (Fig. 1A & B). IVPP V20378 shows typical basal troodontid features such as relatively long forelimbs, elongation of caudal vertebrae, and the third distal phalanges of the second pedal digits forming enlarged claws.

**Cranium.** The skull is three-dimensionally preserved and only missing some of the posterior left elements including the postorbital, parietal, and squamosal, while the rest is well-preserved (Fig. 1A, B). The posterior half of the skull is rotationally deformed to the right of the sagittal plane (Fig. 2).

In lateral view, the skull measures 84 mm in length from the anteriormost extent of the premaxilla to the posterior end of the occipital condyle, with an orbit dorsoventral height of approximately 23 mm. The rostrum is slightly dorsoventrally compressed with the anterior part of the mandible being inserted between the upper jaws. In dorsal view, the sagittal sutures between braincase elements and nasals are clearly visible (Fig. 2A). The right frontal and parietal are well-preserved while the posterior part of left frontal and anterior part of left parietal are missing. The posterior half of the skull is deformed along the right side as the sagittal suture is tilted to the right side but the overall morphology of the right half of the braincase is less impacted taphonomically (Fig. 2C). Since the left side of the braincase is largely missing or distorted, the description and comparative analysis are predominantly based on the right side.

Both Xu et al.[23] and Yin et al.[26] indicated that the foramen magnum is dorsoventrally taller than wide, but the foramen magnum of IVPP V20378 is wider than its dorsoventral height. This difference could be due to intraspecific variation or the result of preservational deformation. A shallow teardrop-shaped subotic recess is preserved on the right side of the braincase but the corresponding area on the left side is not preserved (Fig. 2B). An

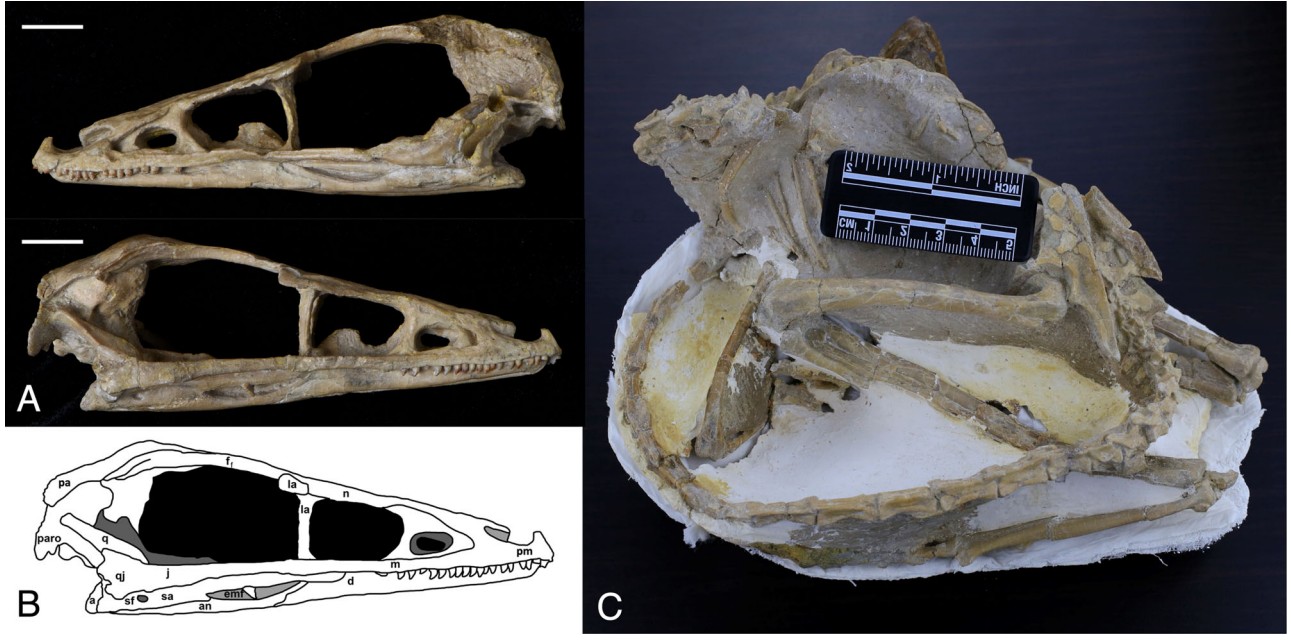

**Fig. 1 The fossil of the troodontid *Sinovenator changii* IVPP V20378. A** photograph of the left lateral view (above) and right lateral view (below) of the skull. **B** corresponding line drawing of the skull in right lateral view. **C** the postcranial skeleton. a articular, an angular, d dentary, emf external mandible fenestra, f frontal, j jugal, la lacrimal, m maxilla, n nasal, pa parietal, paro paroccipital, pm premaxilla, q quadrate, qj quadratojugal, sf surangular foramen, sa surangular. Scale, 1 cm.

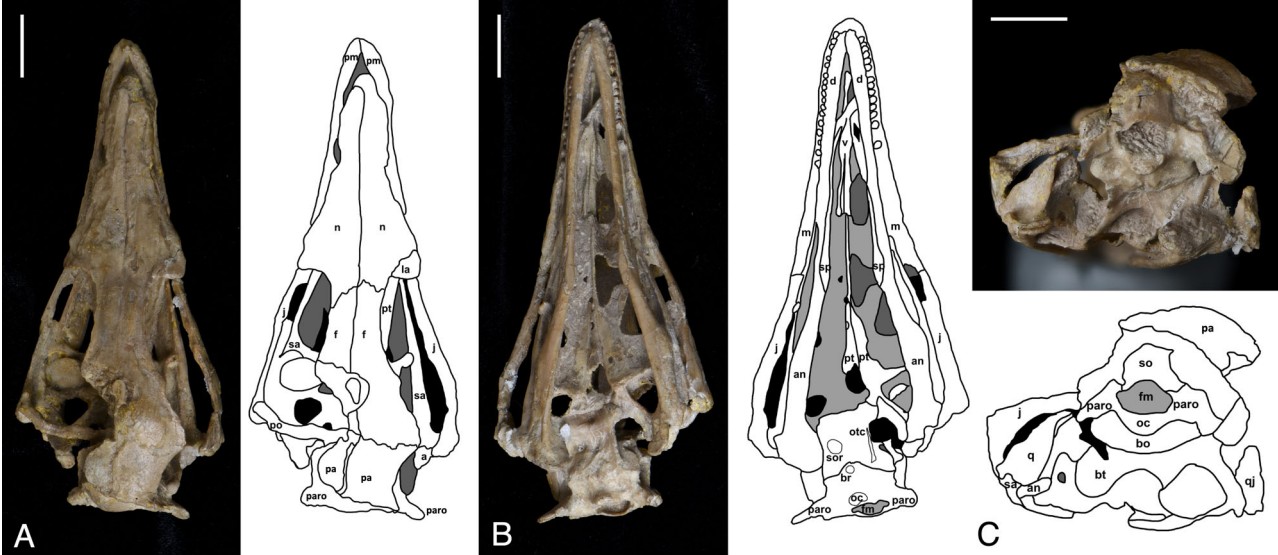

**Fig. 2 The skull of the troodontid *Sinovenator changii* IVPP V20378. A** dorsal view and line drawing. **B** ventral view and line drawing. **C** posterior view and line drawing. a articular, an angular, br basisphenoid recess, bo basioccipital, bt basal tubera, d dentary, f frontal, fm foramen magnum, j jugal, la lacrimal, m maxilla, n nasal, oc occipital condyle, otc otosphenoidal crest, pa parietal, paro paroccipital, po postorbital, pt pteryogoid, q quadrate, qj quadratojugal, sa surangular, so supraoccipital, sor subotic recess, sp splenial, sq squamosal, v vomer. Scale, 1 cm.

even shallower basisphenoid recess than the one found on PMOL-AD00102 is observed in both the ventral view (Fig. 2B), and the otosphenoidal crests on both sides are well preserved. In general, the braincase morphology of IVPP V20378 resembles that of PMOL-AD00102 in having traits that are not present in the holotype, including having a subotic recess, otosphenoidal crest and basisphenoid recess. Although there are differences between the holotype IVPP V 12615, PMOL-AD00102, and IVPP V 20378, they are all assumed to be the same species based on their diagnostic cranial features such as the straight and vertical anterior margin of antorbital fenestra and the T-shaped cross

section of surangular following Xu et al.[23] (Fig. 1 and Supplementary data 3). The straight and vertical anterior margin of the antorbital fenestra, large external naris, densely packed teeth all support a generally intermediate status of *Sinovenator* between other troodontids and other early paravian taxa with respect to craniofacial phenotype.

**Endocranium.** To investigate the neuroanatomical morphology of IVPP V20378, we used the endocast as a proxy for the actual brain surface anatomy (Fig. 3A), based on evidence that the brains of

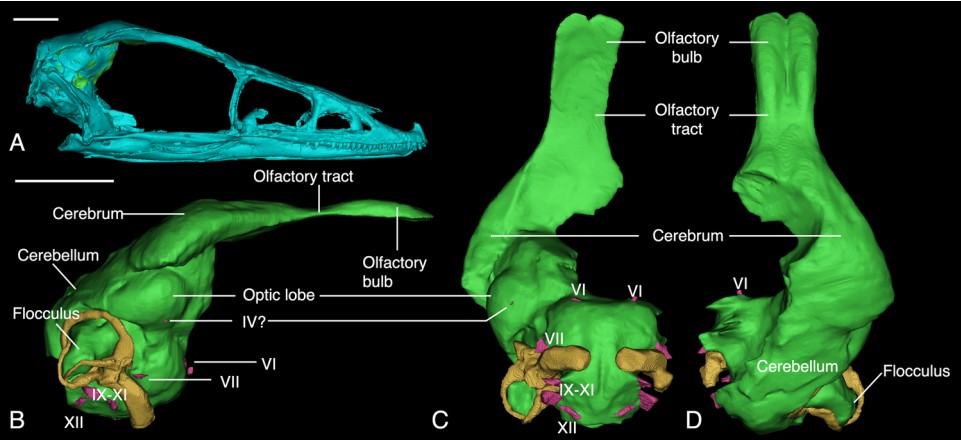

**Fig. 3 3D reconstructions of IVPP V20378 and its cephalic endocast. A**. right lateral view of the skull reconstruction. **B** right lateral view of the endocast, including those of the inner ear (in yellow). **C** and **D** ventral and dorsal views of the endocast. The purple color shows the position of the origin of cranial nerves. Scale, 1 cm.

maniraptorans largely filled the bony endocranial cavity[21,30,31]. The division of neuroanatomical regions follow Balanoff et al.[21]. Both the olfactory bulbs and tracts are proportionately elongated anteriorly and show slight ventral curvature in lateral view. Maniraptorans had a decreasing trend of olfactory capabilities during their evolution, with olfactory bulbs being highly reduced in crown birds, although some bird lineages showed a secondary increase[21,32]. In the case of IVPP V20378, the olfactory bulbs are comparably enlarged relative to crown birds, excluding accipiters and cathartids (Fig. 3). What differentiates *Sinovenator* from extant birds is the presence of a gracile, elongated olfactory tract that connects the olfactory bulbs to the main body of the endocast. The olfactory apparatus (the olfactory bulbs and the olfactory tract) are more reminiscent of the same found in *Archaeopteryx* than that found in modern birds or other troodontids[21]. The large size of the olfactory bulbs and anteriorly long olfactory tracts in *Sinovenator* are plesiomorphic traits for paravians and generally for archosaurs. However, the olfactory apparatus in *Sinovenator* is considerably smaller than in large theropods such as *Tyrannosaurus* or *Allosaurus*, but resembles *Deinonychus*[31] and bears more ventral curvature from lateral view (Fig. 3B). However, the boundary between the olfactory bulb and tract is not clearly defined like other paravians such as *Zanabazar*[21] and *Latenivenatrix*[22,31,33].

The cerebrum is dorsoventrally flat in lateral view but transversely wide (Fig. 3B-D). Posteriorly, the cerebrum is wide, had there not been taphonomic damage, and would have formed a triangular outline with the apex tapering anteriorly to meet the olfactory apparatus (Fig. 3D & 4B). Based on other paravian endocrania, including that of *Zanabazar*[21] and *Latenivenatrixs*[31,33], the flat configuration of the cerebrum is possibly a result from preservational deformation in *Sinovenator*, which also likely caused the midbrain shifting to the right. However, the apparent flatness of many maniraptoran cerebra is probably an artifact of the fact that only the dorsal portion of the cerebrum is recorded in the bony fossa on the frontal whereas the ventral portion is not walled by bone and thus the full dimensions of the cerebrum are not accurately reflected by preserved endocranial surfaces. Anteriorly, a sulcus can be found medially in dorsal view near the junction of the olfactory tract and the cerebrum (Fig. 3D). This interhemispherical sulcus separates two clearly raised left and right sections, although the missing section of the braincase obscures the sulcus. The most posterior point of the sulcus at the boundary of the forebrain and midbrain is visible. The medial and left lateral parts of the cerebrum are not preserved in IVPP V20378, thus we cannot know whether the two cerebral hemispheres are clearly defined by the interhemispherical sulcus or not, but they are in other troodontid endocasts.

The optic lobe increased in proportional size drastically during the evolution of maniraptoran theropods, thus implying greater emphasis on the sense of vision[21,34]. The optic lobes are not clearly defined in most non-maniraptoran theropods, for example, *Majungasaurus*[35] and *Struthiomimus*[31]. Maniraptorans have larger and more clearly defined optic lobes relative to the rest of the brain, and the optic lobes gradually move ventrolaterally to form a more globular brain with modern birds having their optic lobes positioned ventrally to ventroposteriorly relative to the cerebrum in lateral view[21,36]. Consistent with this trend, the right optic lobe of *Sinovenator* is well-defined in both lateral and ventral view (Fig. 3B). In caudal view, the preserved optic lobe of IVPP V20378 is positioned more laterally than in more basal archosaurs[37] and ventral to the posterior section of the cerebrum. The position of the optic lobes in IVPP V20378 is intermediate between non-avialan theropods such as oviraptorosaurians and extant birds and is comparable to that of *Archaeopteryx*[21,38].

The posterior part of the braincase has a misalignment between the skull roof and the supraoccipital, artifactually introducing space between the parietal and supraoccipital, and the squamosal is also detached from the parietal (Fig. 4A). Moreover, given that venous sinuses that dorsally overlie the hindbrain dorsally are widespread but have inconsistent morphology in extant archosaurs[39,40], the hindbrain of *Sinovenator* was reconstructed using other paravian models as references (e.g. *Velociraptor* IGM 100/976[41] and *Zanabazar* IGM 100/1 [[42] and known to lack some correspondence between the external surface of neural tissue and endocranial morphology[40]. The cerebellum in *Sinovenator* is less laterally expanded than both the optic lobe and cerebral area. However, unlike the condition in the troodontid *Zanabazar*[21,34] and the dromaeosaurid *Velociraptor*[41] in which the cerebellum exhibit expanded volume, the cerebellum in *Sinovenator* is flatter dorsoventrally and lacks a well-defined boundary between other endocranial regions (Fig. 3B&D). When viewed laterally, *Sinovenator* has its cerebellum located posterodorsally to the optic lobe and is separated from the cerebrum due to a presence of a mediolateral sulcus (Fig. 3B). The cerebellum in *Sinovenator* occupies a position that is more similar to that of *Archaeopteryx* and extant birds than to the previously mentioned maniraptorans but *Sinovenator* has smaller volume.

As reported previously [28], the acquisition of avian brain architecture may be better described in a mosaic manner than a

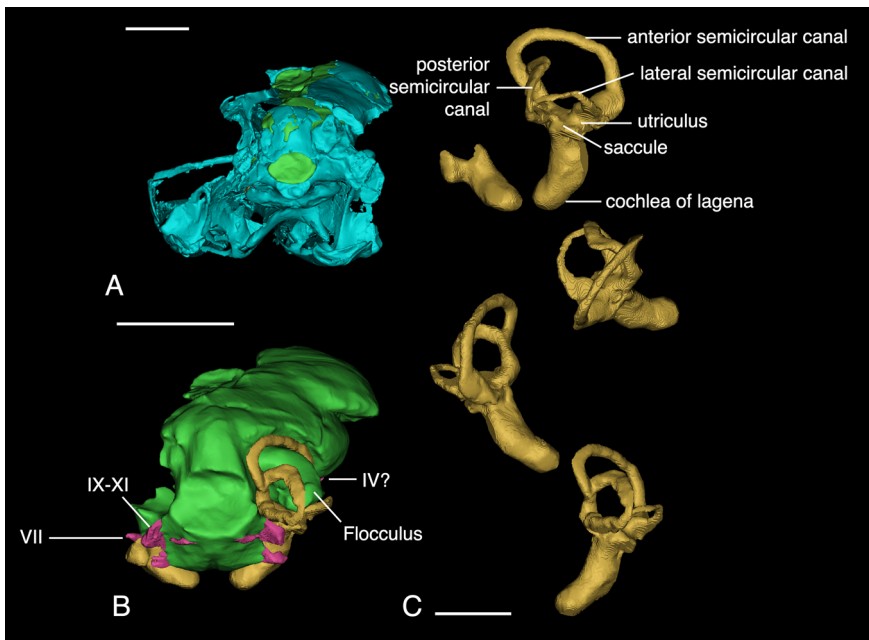

**Fig. 4 3D reconstructions of IVPP V20378 and its cranial endocast (continued) and endosseous labyrinth. A** posterior view of the skull, green shows the position of endocranial cast. **B** posterior view of the brain. **C** from top to bottom, lateral, dorsal, anterior, and posterior view of the right labyrinth. Scale, 1 cm in **A**, **B**, 500 μm in **C**.

gradually linear process. The volumetric expansion and anterior shift of the cerebellum are at least decoupled in the lineage of troodontids. Based on the better-preserved right side of the fossil, the flocculus extends posterior-ventrally with a slight curvature, having a total length of approximately 6 mm. The distal end of the flocculus is aligned with the lateral extent of the optic lobe in the ventral view (Fig. 3C) and adjacent to the posterior extent of the optic lobe in the lateral view (Fig. 3B). Although the floccular endocast is robust, the actual size may not be nearly as large as the endocast indicates[43,44]. Its general morphology resembles the dromaeosaurid *Velociraptor*[41], suggesting a comparable level of visual tracking movements of the eyes, head, and neck. The brainstem is directed ventrally to the optic lobe, indicating a strong dorsoventral flexure contrasting with the more linear profile in non-maniraptoran theropods. The cephalic and pontine angles of IVPP V20378 measure 125° and 122°, respectively; these values resemble those of adult alligator more than those of extant birds[37]. Posteriorly, the foramen magnum creates an oval outline with its major axis extending horizontally. Cranial nerves IX-XI are present along the ventral margin of the cerebellar area (Fig. 3B) though they are mostly unremarkable. A ventral midline fissure is clearly present in the brain stem in ventral view, which is common for modern birds but has not been widely reported in non-avian dinosaurs.

**Endosseous labyrinth.** Only the right endosseous labyrinth is well preserved in IVPP V20378 with the left one missing the vestibular portion of the labyrinth due to poor preservation of this region of the braincase. Broadly, the inner ear morphology resembles that of basal avialans in having a dorsoventrally tall anterior semicircular canal and relatively robust lagena (Fig. 4C[45,46],). The anterior semicircular canal is generally arcuate, contrasting with the triangular anterior semicircular canal in *Velociraptor*[41] but resembling other troodontids (e.g., *Byronosaurus* and IGM 100/3500) and oviraptorosaurians such as *Citipati*[46]. All of the semicircular canals are approximately orthogonal to each other. The posterior semicircular canal bends anteriorly prior to its dorsal connection with the crus communis

(Fig. 4C). The lateral semicircular canal bows dorsally near its midpoint between the ampulla of the lateral semicircular canal and the posterior semicircular canal.

The semicircular canals of the endosseous labyrinth maintain an even lumen thickness throughout most of their extent although none of the canals share the same thickness. More specifically, the anterior semicircular canal is the thickest canal with the posterior semicircular canal being smaller and the lateral semicircular canal being smaller still. All of the canals widen at their respective ampullae adjacent to the vestibule. The ampulla of the posterior semicircular canal is the thickest and the ampulla of the lateral semicircular canal are dorsoventrally compressed. The ampulla of the anterior semicircular canal is the thickest. Considering the intact preservation of the caudal part of the skull and the almost orthogonal direction of ampullae, the differences in their diameters are likely genuine.

The cochlear duct projects ventrally from the vestibule and deflects medially under the brainstem as in other troodontids and the alvarezsauroid *Shuvuuia*[46]. When compared to the vestibular portion of the endosseous labyrinth, the cochlear duct is approximately the same length with no evidence of twisting or further curvature beyond its medial deflection. The cochlear duct is thicker than the case in *Velociraptor*[41] and is slightly more robust than other troodontids[46]. Both the left and right cochlear ducts are similar in size, shape, and angle. Caudally, the fenestra vestibuli and fenestra cochleae are not visible but do not seem to have created a visible impression.

**Geometric morphometric analysis.** Prior to the collection of landmark data from *Sinovenator*, the endocast was digitally mirrored to enable the collection of artificially 'left' sided landmarks to match the landmark scheme of previous studies, then subsequently retrodeformed using median and several pairs of bilaterally symmetric landmarks. While the retrodeformed virtual endocast still exhibited some deformation, we conducted shape analysis on clearly identifiable landmarks that characterize the overall configuration of the neuroanatomical regions. Based on 13 anatomically defined landmarks (see Methods), the non-avialan

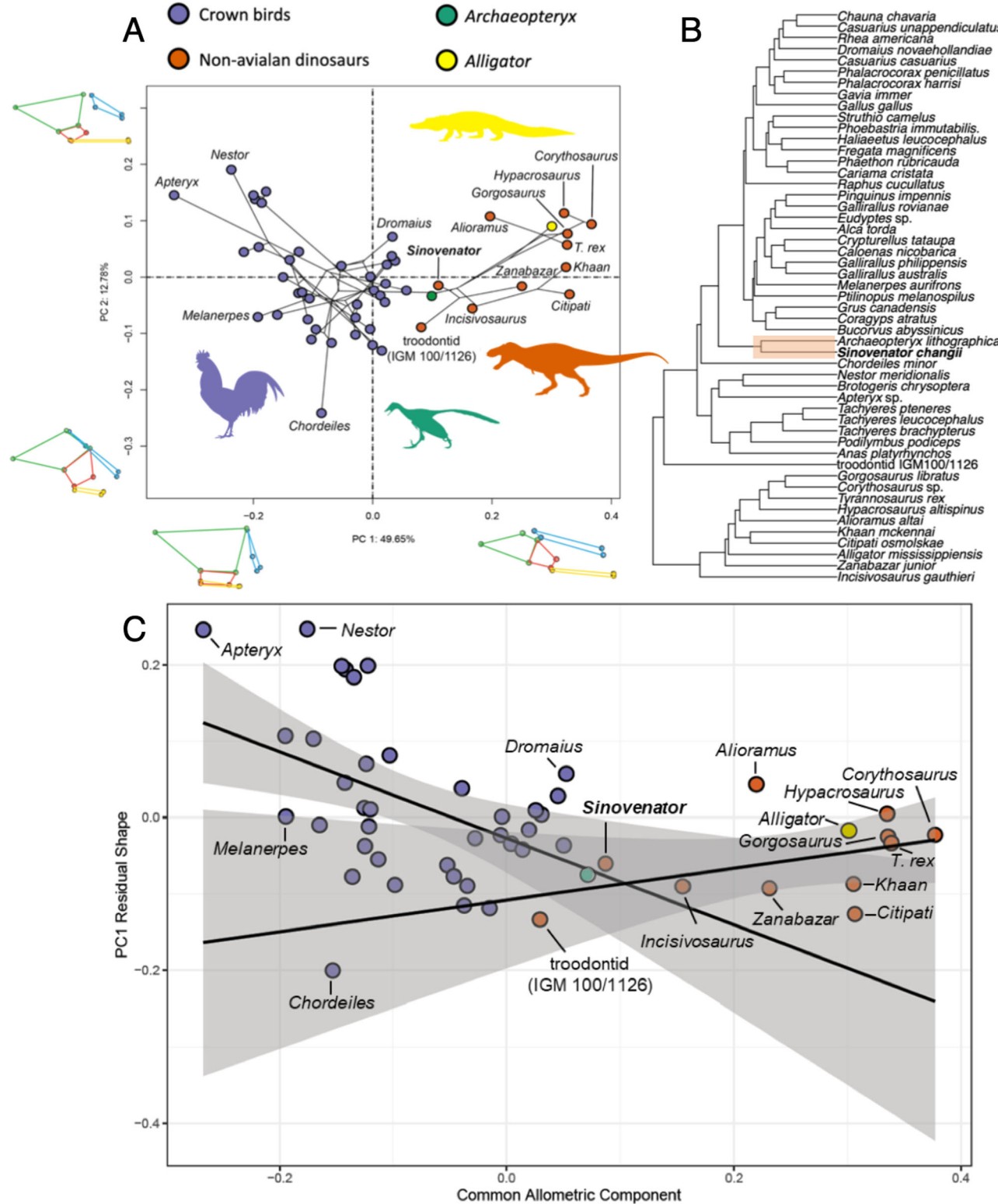

**Fig. 5 Geometric morphometric analysis of *Sinovenator* brain endocast. A** Phylomorphospace based on the first two principal components (PC) of endocast shapes. Insert line images along the axes depict endocranial shapes at extremes of PC1 and 2 axes, colored according to region (green, cerebrum; red, optic lobe; blue, cerebellum; yellow, brainstem). Colored silhouettes of representative taxa are from https://www.phylopic.org/ and under the CC0 1.0 Universal Public Domain Dedication license, *Alligator missisipensis* by Ferran Sayol, *Archaeopteryx lithographica* by Scott Hartman, *T.rex* by Manuel Brea Lueiro, and *Gallus gallus* by Steven Traver. **B** UPGMA clustering analysis based on pairwise Euclidean distances among specimens showing that *Sinovenator* closely resembles *Archaeopteryx*. **C** Bivariate plots of PC1 of residuals from the common allometric component (CAC) against scores along CAC. The solid lines and grey bands indicate regression lines and 95% confidence interval for extant birds and nonavialan dinosaurs separately.

dinosaurs and extant birds cluster separate along the first principal component (PC) axis in the morphospace (Fig. 5A). The first two PC axes account for 49.65% and 12.78% of total endocranial shape variation, respectively. PC1 is primarily associated with the elongation of the entire endocrania, and both PC1 and PC2 correlate with the relative position of the optic lobe and degree of cephalic (midbrain) and pontine flexion. Notably, *Sinovenator* occupies an intermediate area of the morphospace, most closely resembling the endocranial shape of *Archaeopteryx* among sampled taxa, whereas most of the sampled non-avialan theropods cluster around *Alligator* along PC1. The other juvenile troodontid specimen (IGM 100/1126[21]) is also similar in shape to the endocranial shape of *Sinovenator* and *Archaeopteryx*, which approach the endocranial shape variation exhibited by extant birds. We also performed a UPGMA clustering analysis based on pairwise Euclidean distances among specimens (Fig. 5B), demonstrating that *Sinovenator* closely resembles *Archaeopteryx* in endocast shape, which are both within a cluster that includes all extant birds. In addition, the troodontid IGM 100/1126 also clusters with crown birds, with other non-avialan dinosaurs forming a distinct cluster exclusive of extant birds.

Given the large disparity in size, we evaluated the effect of size on endocranial morphology. The endocast shape data were subjected to phylogenetic generalized least squares analysis (PGLS) using log-transformed centroid size of endocasts as a size metric. The results demonstrate an evolutionary (phylogenetically corrected) allometric signal ($R^2 = 0.163$; $P = 0.0093$). To visualize the scaling relationship in endocranial shape, we plotted the PC1 of residual from the common allometric trend (RSC1) against the common allometric component (CAC[47]) (Fig. 5C). The CAC can be seen as the general trajectory of allometry as it is estimated from regression on shape against size (log-transformed centroid size of endocasts), which defines the horizontal axis, and the RSC1 is the first principal component of the residuals that represent deviations from the pooled allometric trajectory. The plot graphically shows that the relationship between endocranial shape and size cannot be modeled uniformly across the entire sampling of archosaurs, including between non-avialan and avialan theropods. When allometric trajectories are created for non-avialan theropods and extant birds separately (*Alligator* and *Archaeopteryx* were excluded in the construction of allometric trajectories), the trendlines show divergent allometric patterns. While all non-avialan dinosaurs overlap with extant birds in RSC1 values, the plot indicates that non-avialan maniraptorans (including *Archaeopteryx*) with smaller endocranial sizes align closer with the trendline for extant birds (Fig. 5C), further supporting the intermediate status of *Archaeopteryx* and maniraptoran endocranial shape in the dinosaur-bird transition. Furthermore, the PC morphospace (Fig. 5A) and RSC1-CAC plot (Fig. 5B) resemble each other in the distribution of data points, implying that PC1 is strongly correlated with size ($R^2 = 0.625$; $p$ value < 0.0001).

## Discussion

In this study we report a well-preserved *Sinovenator* skull specimen with an emphasis on its braincase and its cranial endocast. We also compared its endocranial morphology using landmark-based geometric morphometric methods with non-avialan theropod dinosaur taxa, extant birds and American alligator as outgroup. The results of our comparative quantitative analysis indicate an intermediate brain morphology of *Sinovenator*, as one of the most basal and earliest troodontid species, that neuroanatomically falls between non-maniraptoran theropods and extant birds. The braincase anatomy confirms various traits proposed by Yin et al.[26] in addition to or updating the original description by Xu et al.[23]. The existence of a subtle subotic recess and otosphenoidal crest supports the basal troodontid identity of *Sinovenator* and its intermediate morphology between more later-diverging troodontids and other paravian theropods. Although the braincase of *Sinovenator* and its endocast qualitatively show an intermediate morphology between basal theropods and extant birds, there are also several unique features that may further disclose its neurosensory capabilities. For example, the olfactory tract and bulbs are unexpectedly elongated comparing to other troodontids and paravians, which resembles large theropods[31] and may suggest greater olfactory acuity[22]. Such observation conforms to the increasing trend of olfaction among non-avialan theropod dinosaurs[32].

The coexistence of these endocranial traits, such as elongated olfactory tracts and the posteriorly located optic lobe of *Sinovenator* represent plesiomorphic features for archosaurs, including Troodontidae. When qualitatively compared with late-diverging troodontids or dromaeosaurids, we observe heterogeneity in the acquisition of avialan neuroanatomical features, which has also been observed in later branched avialan taxa[48]. For example, the relative position and shape of a bird-like cerebellum are decoupled. While a linearly arranged brain is characteristic of many archosaurs, including more basal dinosaurs, and thus regarded as the plesiomorphic condition, modern birds and many late-diverging maniraptoran dinosaurs have acquired more globular and dorsoventrally flexed brain architecture[48,49]. And it has been shown that the evolution of avialan-like brain appeared multiple times during the evolution of dinosaurs[21,36]. In *Sinovenator*, the anterior parts of the endocast, including the olfactory tracts and bulbs and the cerebrum, are preserved in relatively linear position and are dorsoventrally flat, but the mid- and hindbrain show strong flexure such that the brainstem is located primarily ventrally to the optic lobe. Although this flexion may be accentuated due to taphonomic distortion, the basic morphology is consistent among three reported *Sinovenator* specimens (IVPP V12615[23] PMOL-AD00102;[26] IVPP V20378 this study). This combination of plesiomorphic and derived endocranial traits further support a mosaic evolution of the avialan and non-avialan dinosaurian brain. Mosaic trends in brain evolution have been reported in multiple bird lineages[50,51], and also in the origin of birds and other theropod lineages[3,27].

Geometric morphometric analysis shows that the allometric trajectories of endocranial morphology in non-avialan dinosaurs and modern birds are divergent, with *Archaeopteryx* and troodontids (*Sinovenator* and unnamed IGM 100/1126) marking an intermediary stage in the dinosaur-bird transition. Extant birds and non-avialan theropods show divergent allometric trends between their size and endocranial shape when regression lines are drawn separately for these two groups. In reality, a single, non-linear model could be a more accurate representation of the allometric trend across archosaurs, but we used two separate linear regressions here to graphically depict the differences in allometric patterns exhibited by non-avialan theropods and extant birds, which have been demonstrated to exhibit disparate endocast shapes[52]. These results are concordant with those attained by Watanabe et al.[52] that suggested that troodontids (IGM 100/1126) occupied intermediate position both in respect to total endocranial shape and allometric trajectory.

It is worth noting that the precise ontogenetic stage of IVPP V20378 is currently unknown and there has not been other *Sinovenator* endocast specimen for intraspecific comparison, which has implications for the interpretation of the results presented here. Both *Archaeopteryx* (Erickson et al.[53]) and the troodontid IGM 100/1126 (Erickson et al. 2007) have been considered juveniles or somatically immature. These two taxa are specimens that exhibit endocast shapes that more closely

resembles those of extant birds than other non-avialan dinosaurs. If IVPP V20378 represents a juvenile *Sinovenator*, then its position in both the PCA and CAC morphospaces (Fig. 5A & C) close to these two taxa and extant birds could be largely due to its immature status. This interpretation implies that modern birds are paedomorphic in brain shape relative to non-avialan paravian theropods, including early avialans such as *Archaeopteryx*. Conversely, if IVPP V20378 exhibits a mature brain shape for the taxon, then derived non-avialan theropods closely related to birds already possessed key features of the avian brain form. For example, non-avialan maniraptoran dinosaurs lie within the allometric trendline for crown birds (Fig. 5C), indicating that these non-avialan dinosaurs may have possessed the same or similar brain shape-to-size relationship to crown birds. These two scenarios are not mutually exclusive (e.g., sustained, stepwise paedomorphic changes could underlie brain shape evolution from non-avialan dinosaurs to crown birds), and previous studies have shown that a single paedomorphic trend does not account for shape changes across brain regions[52]. In fact, the endocranial morphology of *Sinovenator* provides additional evidence of stepwise acquisition of avian brain morphology, with a mixture of juvenile characters (i.e. a posteriorly widened cerebrum; large and topologically distinct optic lobes on endocasts; dorsoventrally flexed configuration), co-occurring with ancestrally more mature configuration of the olfactory apparatus (cf. Fig. 4[31]). As such, although it has been shown that paedomorphism plays essential roles in shaping the skull (Bhullar et al.[54]), and brain[48] across archosaurs, the heterochronic signal exhibits mosaicism across endocranial regions[52]. Taken together, the endocast of the basal troodontid *Sinovenator* qualitatively and quantitatively demonstrates a mixture of derived and plesiomorphic features of the archosaur brain, providing insights into the tempo, timing, and mosaicism of brain evolution along the dinosaur-bird transition.

## Methods

**Specimen and Imaging**. IVPP V20378 is a nearly complete *Sinovenator* skeleton in resting position with the tail surrounding the body, and the skull is disarticulated from the postcranium. The skull and postcranial skeleton of this specimen were scanned separately at the Institute of Vertebrate Paleontology and Paleoanthropology, Beijing, China. The scanning was carried out using the 225 kV micro-computerized tomography (developed by the Institute of High Energy Physics, Chinese Academy of Sciences (CAS)) at the Key Laboratory of Vertebrate Evolution and Human Origins, CAS. The specimen was scanned with a beam energy of 100 kV and a flux of 100 mA at a resolution of 61 μm per pixel using a 360° rotation with a step size of 0.5° and an unfiltered aluminum reflection target. The total dataset of 1,536 slices was reconstructed using two-dimensional reconstruction software MOCUPY developed by the Institute of High Energy Physics, CAS. The segmentation and rendering of the digital models were conducted in Mimics 19.0 (Materilise, Belgium). The 3D models of IVPP V20378 skull and endocranium are available at the open data platform Dryad (Supplementary Data 3, https://doi.org/10.5061/dryad.41ns1rnk6) or upon request to the 1st or correspondence author.

**Morphometric Data & Analysis**. The landmark-based geometric morphometric analysis is based on the discrete landmarks and taxonomic sampling used in Gold and Watanabe[55] and Watanabe et al.[52] (Supplementary Data 1 & 2). These landmarks include (1) the anterior-most median point of the cerebrum; (2) posterior-most median point of the cerebrum; (3) dorsal junction point between the cerebrum and optic lobe; (4) ventral junction point

between the cerebrum and optic lobe; (5) lateral junction point between the brainstem and midbrain; (6) junction point between the optic lobe and the border between the cerebellum and brainstem; (7) anterior-most median point of the cerebellum; (8) posterior-most median point of the cerebellum; (9) anterolateral point of the cerebellum; (10) posterolateral point of the cerebellum; (11) anterior-most median point of the brainstem; (12) posterior-most median point of the brainstem; and (13) the ventrolateral point of the brainstem. Unlike previous studies using this landmark scheme, the semi-landmarks were excluded from the landmark scheme for this study due to the concern for unreliable placement of curve and surface semi-landmarks given the damaged and deformed state of the *Sinovenator* endocast. Because the coordinate data from previous studies comprise the median and left-sided landmarks, the virtual endocast of *Sinovenator* was first mirrored in GeoMagic Wrap (3D Systems). Analysis was conducted in RStudio with R statistical language v4.2.1 (R Core Development Team 2022). To retro-deform the endocast, we performed a symmetrization algorithm[56] implemented in the Morpho R package[57] using observable median points and pairs of bilaterally symmetric points. The coordinate data were then subjected to generalized Procrustes analysis minimizing Procrustes distance[58,59], followed by a principal components analysis (PCA) and phylogenetic least squares analysis (PGLS) using the gm.princomp and procD.pgls functions in the 'geomorph' R package v4.0[60]. Allometric trajectories were plotted with the aid of the "cac" function in the "Morpho" R package[57]. For clustering analysis, we generated a UPGMA tree using the "upgma" function in the "phangorn" R package[61].

**Statistics and reproducibility**. Fifty-one taxa and 13 three-dimension landmarks are used in this study. The data are deposited as Supplementary Data 1 & 2 and the R code for this study is deposited as Supplementary Data 4.

**Supplementary data**. All the data mentioned below can be found at the open-access database Dryad at https://doi.org/10.5061/dryad.41ns1rnk6.

1. Phylogenetic tree used for phylomorphospace plotting in geometric morphometric analysis.
2. Coordinate data of brain endocast used in this study.
3. 3D models of the skull, brain endocast, and endosseous labyrinth of IVPP V20378.
4. R code for processing and analyzing the data for this study.

**Reporting summary**. Further information on research design is available in the Nature Portfolio Reporting Summary linked to this article.

## Data availability

The supplementary data can be reached at open access database Dryad at https://doi.org/10.5061/dryad.41ns1rnk6. All data related to this study can also be available upon reasonable request to the corresponding or 1st author.

## Code availability

The code for processing and analyzing the endocranial shape data can be accessed on Dryad (https://doi.org/10.5061/dryad.41ns1rnk6). The code was written in and for R v4.2.1.

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

## Acknowledgements

We thank three anonymous reviewers for their comments that substantially improved this study. We also would like to thank Hailong Zang for photography and Yun Feng for the CT scan of *Sinovenator* (IVPP V20378); Amy Balanoff, Eugenia Gold and Mark Norell for CT imaging and endocranial reconstructions of extant birds and non-avian dinosaurs sampled in this study. The colored silhouettes used in Fig. 5 are done by artists who selflessly make their work public. This work was supported by the National Natural Science Foundation of China (42288201) to X. X.

## Author contributions

C.Y., Z.Q., and X.X. designed the project. Q.M. and X.X. helped prepare the specimen and curated the 3D data from CT scan. C.Y. segmented and created 3D endocranial reconstruction of *Sinovenator* (IVPP V20378). C.Y., A.W., Z.Q. J.L.K., L.M.W., and X.X. identified, compared, and described the anatomy of the skull and reconstructed endocasts. A.W. conducted the geometric morphometric analysis. The manuscript and figures were prepared with everyone's contribution.

## Competing interests

The authors declare no competing interests.
