## [Peer Review File · Communications Biology]

Reviewers' comments:

Reviewer #1 (Remarks to the Author):

The ms by Yu and colleagues performs a 3D CTscan reconstruction of the skull of a specimen of *Sinovenator* from the Institute of Vertebrate Paleontology and Paleoanthropology in Beijing, China. The study focuses in describing the endocranial cavity in more detail than the rest of the skull to fill a gap in the knowledge of brain evolution in this troodontid "transitional" forms (i.e., between troodontids and paravians towards avians). To such end, the CT was segmented and reconstructed and thereafter submitted to a geometric morphometrics comparative setting. While the study is interesting and the material merits its communication to the scientific community, many aspects of the ms need to be taken care of before publication.

The introduction says that this is a morphometrics study while, after reading the anatomical description, it looks like there is way more description than analyses. I wasn't able to finally understand if the results highlight convergence, transition, or even both. Much of this was because the discussion requires consideration; while it is true that there are differences in the arrangement between brain parts in this specimen compared to other avian and non-avian taxa, I find it difficult to link them to "mosaic" evolution. Notice that the brain across crown-birds shows flexed and non-flexed configurations, and so far there is no hypothesis linking this variation to evolution or function. If the authors consider it important they would need to explain why. The functional rationales, such as those relating relative size of functional portions such as the olfactory bulb, need to be better documented and further developed in the discussion to make the case. Regarding GM, there should be depictions of the PCs to see what the resemblance actually means (i.e., proximity in morphospace) and an evaluation of evolutionary allometry could be extremely helpful, given the fact that brain evolution is classically viewed from such standpoint (encephalization). Furthermore, couldn't shape differences (what was interpreted as a mosaic) actually be underlined by allometry? importantly, there are specific analyses that help to evaluate the degree of convergence (in R), and these would be necessary to make sense out of the observed patterns from a GM standpoint (i.e., descriptive). Detailed comments can be found below:

L 30 Lobes

L 46 Tremendous sounds exaggerated

L 76-78 Sentences need revision

L 93 According to who?

L 100 Shows? Who?

L 102 English rev?

L 108-109 This study encompasses a whole description (qualitative) more than any other thing...

L 120 Are there more diagnostic features? Citation?

L 143 I find it hard to link it to variation...actually, the whole cranium is dorsoventrally deformed, affecting not only the forebrain but also the olfactory lobe to some extent (apparently)...couldn't all these features be related to such vector of deformation. Accordingly, why not trying to retrodeform the skull as a whole? Shouldn't be difficult and would render a more accurate and "real-looking" endocranium (and skull).

150-151 type of V in IVPP

L 152 Overall morphology supports it is intermediate by eye?(!)

L 168 Like crocodiles?

L 170 Italics needed

L 195 Occupies

L 196 This is said above ...citation supporting it?

L 210 Introducing

L 214 What models was this reconstruction based on? Why? On what basis?

L 219 If everything was estimated, how can you know? This needs to be developed further
L 221 This position's description seems largely a guess (although it is obvious where the cerebellum is)...is it necessary? Perhaps GM could help here (even if only with a pairwise superimposition)
L 227 Where is the Flocculus?? Can't see without labelling (nor estimate its relative size).
L 232-33 Can you provide labelling plus lines and/or angle? It would nice to show in contrast to others or at least a value of angles (Table perhaps?...anything that allows comparison)
L249 Labels?
L 274 Why not simply duplicate the landmarks of the good side and stich both? I would understand this (and support it indeed!) if the model was then retrodeformed to yield a nice 3D model to print etc., even if it would not serve for landmarking without the proper caveat.
L 279 Morphospace does not separate...forms separate over it. Although crown-bird separation from the rest is clear, the rest is less evident. It would be very important to label who is who near Sinovenator (there is a young troodontid, text says, but who is the other taxon). Also, PCA can be tricky and things that fall nearby might not be that similar. I would strongly suggest depicting a minimum spanning tree on top of the PCA (and the phylogeny –phylomorphospace also —, for instance, or instead, build a UPGA dendrogram to make this clearer.
L 301-304 need citations to discuss and justify
L 306 olfaction?

Figures

1 Labels in skull sketch are too tiny, impossible to read. In Figure 2 the same thing. Please enlarge. Would it be possible to have a 3D render?
3 What is the loos piece in 3E near the canals? Can N. II be guessed? Why not add a dorsal view? That would help visualization of forebrain breadth clearer as well as that of olfactory bulbs. A depiction of the landmarks would be very desirable.
Figure 4 The cladogram is too small. Please consider labelling the taxa near Sinovenator (why not all?). There should be a table somewhere that explain what taxa were included. It is essential to show what PCs mean in terms of shape change and how much variance they explain. Please consider also included the info about phylogeny and real similitude (e.g., MST and/or UPGMA); this is very important

Reviewer #2 (Remarks to the Author):

Since the works of Domínguez Alonso et al. (2004) and Franzosa (2004), it has been pretty clear that Archaeopteryx and some more basal eumaniraptorans had a similar brain morphology. The paucity of data, however, made it unclear if that morphology was due to homoplasy or was inherited from a common ancestor. Sinovenator changii is a troodontid, the neurocranial morphology of which is known thanks to the specimens IVPP V 12615 (figured in right lateral view only in Xu et al., 2002: fig. 1b) and PMOL-AD00102 (studied in detail by Yin et al., 2018). The present study is based on a third specimen, IVPP V20378, which is beautifully preserved. Although the title claims that the brain morphology of Sinovenator is convergent with that of avialans, I fail to see any demonstration of homoplasy here. The mere fact that the brain of Sinovenator closely resembles that of Archaeopteryx is not sufficient to sustain a claim of convergent evolution. I don't understand very well either the few lines of discussion about the "mosaic pattern". The geometric morphometric analysis is very interesting, but the results are discussed rather too quickly: the discussion should be more elaborated and detailed.

I list below some suggestions for improvement:

Li. 64: this is a recurring misunderstanding. Rogers (1998) did not reconstruct any digital endocast. Instead, he directly scanned a natural endocast. The first digital reconstruction of a dinosaur endocast

made from the scanning of a braincase is due to Knoll et al. (1999).

Knoll, F., E. Buffetaut & M. Bülöw. 1999. A theropod braincase from the Jurassic of the Vaches Noires cliffs (Normandy, France): osteology and palaeoneurology. *Bulletin de la Société géologique de France*, 170 (1): 103–109.

Li. 107: What do you mean “the reconstruction of the cranial endocast is limited”? Yin et al. (2002) did not provide any endocast reconstruction, limited or not.

Li. 127 et seq.: caudal & rostral would be better, in my opinion, than posterior & anterior.

Li. 132: “the anterior part of the mandible being largely inserted in between the upper jaws”: I don’t see anything unusual here in your photos.

Li. 152: “and the overall morphology supports an intermediate status of Sinovenator between other troodontids and other early paravian taxa”. Please, elaborate.

Li. 159: you could cite Knoll & Kawabe (2020) here:

Knoll, F., & S. Kawabe. 2020. Avian palaeoneurology: reflections on the eve of its 200th anniversary. *Journal of Anatomy*, 236 (6): 965–979.

Li. 162: “slight ventral curvature”. The olfactory bulbs are better described as dorsally convex in lateral view.

Li. 166: “relatively large” is better than “comparably enlarged”.

Li. 168: you can drop “via the cerebrum”.

Li. 171: “anteriorly long olfactory tracts”: the tracts are very short in Sinovenator.

Li. 176: if you do mean Zanabazar, the reference you want to cite is Norell et al. (2009). However, the olfactory tracts look pretty much the same in Zanabazar and Sinovenator.

Li. 180: “along the coronal axis”: Do you simply mean “transversally”?

Li. 183: Troodon or Latenivenatrix? Be consistent.

Li. 186: “not being walled by bone”: Do you mean in vivo or as preserved?

Li. 194: “The optic lobe has increased in size significantly during the evolution of maniraptorans”: Any reference to back this up?

Li. 201: You can cite Walsh & Knoll (2018) here.

Walsh, S. A., & F. Knoll. 2018. The evolution of avian intelligence and sensory capabilities: the fossil evidence. *Digital Endocasts: from Skulls to Brains*. 59–69. Eds: E. Bruner, N. Ogiwara & H. C. Tanabe. Tokyo: Springer.

Li. 203: How could you tell, in lateral view, that a structure is positioned more laterally?

Li. 214: Well, this is problematic. I think you should at least provide a figure of the original endocast (that is, before such amendments are made).

Li. 241: the labyrinth looks derived with respect to those of oviraptorosaurs, dromaeosaurids and some other troodontids (see e.g., Hanson et al., 2021).

Hanson M, Hoffman EA, Norell MA, Bhullar BS. The early origin of a birdlike inner ear and the evolution of dinosaurian movement and vocalization. *Science*, 372(6542): 601-609.

I beg to disagree about its resemblance with that of Velociraptor. In the latter, the RSCC is relatively much shorter, whereas in Sinovenator it is relatively much longer and arcuate. Thus, it extends dorsally much beyond the apex of the crus commune in Sinovenator, but not in Velociraptor. In this respect, the labyrinth of Sinovenator looks more bird-like. In lateral view, the vestibular system of the inner ear of Sinovenator does not look triangular at all to me. The lagena is remarkably robust, much more so than in Velociraptor.

Li. 255: Could that be due to preservational issues?

Li. 259: “though this may be due...”: I don’t understand.

Li. 264: “with no evidence of twisting...”: I disagree, it is apparent that the lagena not only curves medially but also caudally.

Li. 265: I disagree. See above.

Li. 266: “Laterally...”. Please, rephrase.

Li. 314: You can cite Walsh & Knoll (2018) here as well.

Li. 317: tract, not tact.

Fig.: please provide dorsal and ventral views of the endocast. Also label each cranial nerve clearly.

Reviewer #3 (Remarks to the Author):

Xu et al. describe the endocast of *Sinovenator* based on an exceptionally well-preserved skull belonging to this taxon. The authors interpret the similarity between the endocast of *Sinovenator* and *Archaeopteryx* as the result of convergence among paravian dinosaurs. Overall, I enjoyed the manuscript, which is well written and executed. I have only few major comments regarding the main interpretation of the findings, additional analyses, and sharing of the data.

Major comments:

- General interpretation of brain evolution: can the authors explain why this should be convergence and not being representative of the plesiomorphic condition for paravians? We don't currently have any endocast regarding small-bodied or early diverging dromaeosaurs (not published at least), exception made for *Bambiraptor* which is considered a juvenile. Maybe it is only me, but I could perfectly picture something like the endocast of *Sinovenator* or *Archaeopteryx* as an early paravian condition, rather than convergence among relatively closely related avialan and non-avian dinosaur. The authors kind of suggest this in the abstract, but I would love to hear more about the evolutionary scenario that they depict based on their novel results. I would also suggest the authors to run an evolutionary rate analyses using the PCoordinates and the assembled informal supertree to see if indeed this is a matter of convergence or not based on current data.
- Geometric morphometrics: I believe that the initial part of the paragraph in the results can be moved to the materials and methods. On the other hand, a sentence giving an idea of the number of taxa and their phylogenetic attribution might be useful here, considering that, without checking the figures, it is currently impossible to grasp an overview of the comparative dataset and sample size. Additionally, I find the overall description of what goes closer to what in the morphospace quite preliminary for the overall goal of the study (demonstrating convergence/morphological similarity between the endocast of early avialans and non-avian dinosaurs). I would require the authors to run a cluster analysis to actually evaluate if *Archaeopteryx*, *Sinovenator*, and the young troodontid fall within one single cluster or not.
- Data policy: Communication biology embraces open access. Yet there is no statement regarding how the CT scans and endocast volume will be treated after publication. I would kindly ask the authors to make at least the .stl files of the skull and endocast (including semicircular canal) freely available upon publication of the manuscript.
- Figure 4A is impossible to read. Please, make sure to improve the resolution of the figure before publication
- I believe that important references are still missing from the manuscript, especially for comparative purposes. I would ask the authors to include the following, that I am sure they will find useful to expand their discussion and comparative framework

References:

- 1) Alonso, P. D., Milner, A. C., Ketcham, R. A., Cookson, M. J., & Rowe, T. B. (2004). The avian nature of the brain and inner ear of *Archaeopteryx*. *Nature*, 430(7000), 666-669.
- 2) Beyrand, V., Voeten, D. F., Bureš, S., Fernandez, V., Janáček, J., Jirák, D., ... & Tafforeau, P. (2019). Multiphase progenetic development shaped the brain of flying archosaurs. *Scientific Reports*, 9(1), 10807.
- 3) Hanson, M., Hoffman, E. A., Norell, M. A., & Bhullar, B. A. S. (2021). The early origin of a birdlike inner ear and the evolution of dinosaurian movement and vocalization. *Science*, 372(6542), 601-609.
- 4) Fabbri, M., Mongiardino Koch, N., Pritchard, A. C., Hanson, M., Hoffman, E., Bever, G. S., ... & Bhullar, B. A. S. (2017). The skull roof tracks the brain during the evolution and development of reptiles including birds. *Nature ecology & evolution*, 1(10), 1543-1550.

5) Ksepka, D. T., Balanoff, A. M., Smith, N. A., Bever, G. S., Bhullar, B. A. S., Bourdon, E., ... & Smaers, J. B. (2020). Tempo and pattern of avian brain size evolution. *Current Biology*, 30(11), 2026-2036.

Minor comments:

- Line 93: the authors should make clear if these features were the original list recovered by Xu et al. (2002) or not. It is not clear at the moment
- The last paragraph of the introduction would be better suited as the second to last: the last paragraph sounds as a recapitulation of the state of the art regarding the cranial morphology of the investigated taxon. For this reason, I believe that the paragraph starting with "In this study, we..." should be placed at the end of the introduction
- Line 143: what is the correct shape of the foramen magnum according to the authors' view and experience with these fossils? A clear indication regarding this matter is needed for future reference

Referee expertise:

Referee #1: Avian palaeoneurology

Referee #2: Avian palaeoneurology

Referee #3: Avian palaeoneurology

Reviewers' comments:

General replies:

1. The manuscript and other submitted materials have been revised according to comments from three reviewers.
2. Revised figures for better illustration of the endocast anatomy, and we deposit the 3D reconstruction of the skull and brain endocast to open data platform Dryad (<https://doi.org/10.5061/dryad.41ns1rnk6>).
3. We replace the convergent claim in both the title and main text with avialan-like. And on the basis of more detailed geometric morphometric analysis, we show that the brain endocast of *Sinovenator* had many avialan features and probably represent transitional status of brain development according to the allometric plotting.

Reviewer #1 (Remarks to the Author):

The ms by Yu and colleagues performs a 3D CTscan reconstruction of the skull of a specimen of *Sinovenator* from the Institute of Vertebrate Paleontology and Paleoanthropology in Beijing, China. The study focuses in describing the endocranial cavity in more detail than the rest of the skull to fill a gap in the knowledge of brain evolution in this troodontid “transitional” forms (i.e., between troodontids and paravians towards avians). To such end, the CT was segmented and reconstructed and thereafter submitted to a geometric morphometrics comparative setting. While the study is interesting and the material merits its communication to the scientific community, many aspects of the ms need to be taken care of before publication.

The introduction says that this is a morphometrics study while, after reading the anatomical description, it looks like there is way more description than analyses. I wasn't able to finally understand if the results highlight convergence, transition, or even both. Much of this was because the discussion requires consideration; while it is true that there are differences in the arrangement between brain parts in this specimen compared to other avian and non-avian taxa, I find it difficult to link them to “mosaic” evolution. Notice that the brain across crown-birds shows flexed and non-flexed configurations, and so far there is no hypothesis linking this variation to evolution or function. If the authors consider it important they would need to explain why. The functional rationales, such as those relating relative size of functional portions such as the olfactory bulb, need to be better documented and further developed in the discussion to make the case. Regarding GM, there should be depictions of the PCs to see what the resemblance actually means (i.e., proximity in morphospace) and an evaluation of evolutionary allometry could be extremely helpful, given the fact that brain evolution is classically viewed from such standpoint (encephalization). Furthermore, couldn't shape differences (what was interpreted as a mosaic) actually be underlined by allometry? importantly, there are specific analyses that help to evaluate the degree of convergence (in R), and these would be necessary to make sense out of the observed patterns from a GM standpoint (i.e., descriptive). Detailed comments can be found below:

L 30 Lobes
L 30 corrected
L 46 Tremendous sounds exaggerated
L 46 tremendous efforts → considerable progress
L 76-78 Sentences need revision
L 76-78 sentence structures revised
L 93 According to who?
L 93 Xu et al. (2002) added in the end of sentence
L 100 Shows? Who?
L 100 shows→show, by Yin et al. (2018) added in the end of sentence
L 102 English rev?
L 102 sentence structures revised
L 108-109 This study encompasses a whole description (qualitative) more than any other thing...
L 108-109 Yes, Yin et al. (2018) <https://peerj.com/articles/4977/> is mostly morphological description and a short phylogeny reconstruction.
L 120 Are there more diagnostic features? Citation?
L 120 diagnostic features are listed in revised manuscript line 94-97.
L 143 I find it hard to link it to variation...actually, the whole cranium is dorsoventrally deformed, affecting not only the forebrain but also the olfactory lobe to some extent (apparently)...couldn't all these features be related to such vector of deformation. Accordingly, why not trying to retrodeform the skull as a whole? Shouldn't be difficult and would render a more accurate and "real-looking" endocast (and skull).
L 143 We mention the possibility that variation of the foramen magnum can be due to deformation or variation. The brain endocast only occupies a small region of the whole skull, thus we chose to retrodeform the reconstructed endocast rather than the whole skull. The dataset we used for GM analysis were only retrodeformed the endocranium rather than the skull (for example Watanabe et al. 2021 *elife*), thus we followed the same workflow.
L 150-151 type of V in IVPP
L 150-151 the additional space is deleted. IVPP Vnumber stands for Vertebrate specimens collected in IVPP.
L 152 Overall morphology supports it is intermediate by eye?(!)
L 152 the sentence is rephrased and more detailed comparisons (qualitative and quantitative) are in following sections
L 168 Like crocodiles?
L 168 could say *Sinovenator* olfactory tracts are more like crocodiles than modern birds, but are still wider than crocodiles (illustrated by Jirak & Janacek 2017). They are most similar to other basal theropods.
L 170 Italics needed
L 170 Archaeopteryx → *Archaeopteryx*
L 195 Occupies
L 195 occupis → occupies
L 196 This is said above ...citation supporting it?
L 196 repeated sentence on the increased optic lobe deleted. References added.
L 210 Introducing
L 210 ntroducing → introducing
L 214 What models was this reconstruction based on? Why? On what basis?
L 214 reference model added, *Velociraptor* IGM 100/976 King et al. (2020)
L 219 If everything was estimated, how can you know? This needs to be developed further

L 219 References added with more quantitative measurements regarding the volumetric expansion
L 221 This position's description seems largely a guess (although it is obvious where the cerebellum is)...is it necessary? Perhaps GM could help here (even if only with a pairwise superimposition)
L 221 the position description deleted.
L 227 Where is the Flocculus?? Can't see without labelling (nor estimate its relative size).
L 227 the flocculus is labeled in Fig. 3D
L 232-33 Can you provide labelling plus lines and/or angle? It would nice to show in contrast to others or at least a value of angles (Table perhaps?...anything that allows comparison)
L 232-33 we added the cephalic and flexure angles measurements and made more discussion in later sections
L249 Labels?
L 249 Fig. 3E label is added.
L 274 Why not simply duplicate the landmarks of the good side and stitch both? I would understand this (and support it indeed!) if the model was then retrodeformed to yield a nice 3D model to print etc., even if it would not serve for landmarking without the proper caveat.
L 274 to keep consistence we follow the workflow from Watanabe et al. (2021) as the comment from line 143 above (although I am not sure I really understand this)
L 279 Morphospace does not separate...forms separate over it. Although crown-bird separation from the rest is clear, the rest is less evident. It would be very important to label who is who near *Sinovenator* (there is a young troodontid, text says, but who is the other taxon). Also, PCA can be tricky and things that fall nearby might not be that similar. I would strongly suggest depicting a minimum spanning tree on top of the PCA (and the phylogeny –phyломorphospace also —, for instance, or instead, build a UPGA dendrogram to make this clearer.
L 279 the subject of “separates” is changed to taxon clusters, label other taxa near *Sinovenator*, a minimum spanning tree has been added in Figures.
L 301-304 need citations to discuss and justify
L 301-304 we deleted the claim about the hypothesized insectivory feeding habits. The sentence about olfaction is rephrased and two references are added
L 306 olfaction?
L 306 olfaction → olfactory tracts

Figures

1 Labels in skull sketch are too tiny, impossible to read. In Figure 2 the same thing. Please enlarge. Would it be possible to have a 3D render?
The labels in figure 1 and 2 are enlarges and missing labels (as mentioned in L 249 comment) are added. 3D renders of the entire brain, skull, and inner ears are deposited in open data platform Dryad.
3 What is the loos piece in 3E near the canals? Can N. II be guessed? Why not add a dorsal view? That would help visualization of forebrain breadth clearer as well as that of olfactory bulbs. A depiction of the landmarks would be very desirable.
More clear labels and a dorsal view have been added. Landmarks are provided in supplementary materials.
Figure 4 The cladogram is too small. Please consider labelling the taxa near *Sinovenator* (why not all?). There should be a table somewhere that explain what taxa were included. It is essential to show what PCs mean in terms of shape change and how much variance they explain. Please consider also included the info about phylogeny and real similitude (e.g., MST and/or UPGMA); this is very important
A minimal spanning tree (MST) is added to replace the original tree, and a .tre file is provided in supplementary materials.

Reviewer #2 (Remarks to the Author):

Since the works of Domínguez Alonso et al. (2004) and Franzosa (2004), it has been pretty clear that Archaeopteryx and some more basal eumaniraptorans had a similar brain morphology. The paucity of data, however, made it unclear if that morphology was due to homoplasy or was inherited from a common ancestor. *Sinovenator changii* is a troodontid, the neurocranial morphology of which is known thanks to the specimens IVPP V 12615 (figured in right lateral view only in Xu et al., 2002: fig. 1b) and PMOL-AD00102 (studied in detail by Yin et al., 2018). The present study is based on a third specimen, IVPP V20378, which is beautifully preserved. Although the title claims that the brain morphology of *Sinovenator* is convergent with that of avialans, I fail to see any demonstration of homoplasy here. The mere fact that the brain of *Sinovenator* closely resembles that of *Archaeopteryx* is not sufficient to sustain a claim of convergent evolution. I don't understand very well either the few lines of discussion about the "mosaic pattern". The geometric morphometric analysis is very interesting, but the results are discussed rather too quickly: the discussion should be more elaborated and detailed. See general reply 3 and also below. More quantitative analyses are added and discussion section is largely restructured.

I list below some suggestions for improvement:

- Li. 64: this is a recurring misunderstanding. Rogers (1998) did not reconstruct any digital endocast. Instead, he directly scanned a natural endocast. The first digital reconstruction of a dinosaur endocast made from the scanning of a braincase is due to Knoll et al. (1999).
Knoll, F., E. Buffetaut & M. Bülow. 1999. A theropod braincase from the Jurassic of the Vaches Noires cliffs (Normandy, France): osteology and palaeoneurology. *Bulletin de la Société géologique de France*, 170 (1): 103–109.
- Li. 64 the statement is corrected and Knoll et al. (1999) is added in reference.
- Li. 107: What do you mean "the reconstruction of the cranial endocast is limited"? Yin et al. (2002) did not provide any endocast reconstruction, limited or not.
- Li. 107 Yin et al. (2018) didn't reconstruct the cranial endocast, only has a paragraph on the braincase morphology in page 22 and Figure 11. The sentence is rephrased.
- Li. 127 et seq.: caudal & rostral would be better, in my opinion, than posterior & anterior.
- Li 127 the anatomical terminologies used here are following Xu et al. (2002) and Yin et al. (2018), thus anterior-posterior are used mostly rather than rostral-caudal.
- Li. 132: "the anterior part of the mandible being largely inserted in between the upper jaws": I don't see anything unusual here in your photos. Li 132 the insertion means over occlusion of upper and lower jaws, it is clearer from the skull 3D rendering.
- Li. 152: "and the overall morphology supports an intermediate status of *Sinovenator* between other troodontids and other early paravian taxa". Please, elaborate.
- Li 152 we added more detail about the skull of *Sinovenator*
- Li. 159: you could cite Knoll & Kawabe (2020) here:
Knoll, F., & S. Kawabe. 2020. Avian palaeoneurology: reflections on the eve of its 200th anniversary. *Journal of Anatomy*, 236 (6): 965–979.
- Li 159 Knoll & Kawabe (2020) reference added
- Li. 162: "slight ventral curvature". The olfactory bulbs are better described as dorsally convex in lateral view.
- Li 162 ventral curvature from lateral view → dorsally convex in lateral view
- Li. 166: "relatively large" is better than "comparably enlarged".
- Li 166 comparably enlarged → relatively large
- Li. 168: you can drop "via the cerebrum".
- Li 168 "via the cerebrum" deleted
- Li. 171: "anteriorly long olfactory tracts": the tracts are very short in *Sinovenator*.

Li 171 *Sinovenator* olfactory tracts are longer than many non-paravian theropods as shown by Balanoff et al. (2013)

Li. 176: if you do mean *Zanabazar*, the reference you want to cite is Norell et al. (2009). However, the olfactory tracts look pretty much the same in *Zanabazar* and *Sinovenator*.

Li 176 Norell et al. (2009) *A Review of the Mongolian Cretaceous Dinosaur Saurornithoides* (Troodontidae: Theropoda) presented detailed description of *Zanabazar* including its braincase and endocast (e.g. Fig. 24-28). And more comparisons between *Zanabazar* and *Sinovenator* are added here and elsewhere.

Li. 180: “along the coronal axis”: Do you simply mean “transversally”?

Li 180 yes, along the coronal axis → transversally

Li. 183: Troodon or *Latenivenatrix*? Be consistent.

Li 183 *Troodon* → *Latenivenatrix*s, and added reference Van Der Reest & Currie (2017) for the naming of *Latenivenatrix*, on the basis of specimen TMP 79.8.1 and others.

Li. 186: “not being walled by bone”: Do you mean in vivo or as preserved?

Li 186 *in vivo*

Li. 194: “The optic lobe has increased in size significantly during the evolution of maniraptorans”: Any reference to back this up?

Li 194 references added Balanoff et al. (2013) and Torres et al. (2021)

Li. 201: You can cite Walsh & Knoll (2018) here.

Walsh, S. A., & F. Knoll. 2018. The evolution of avian intelligence and sensory capabilities: the fossil evidence. *Digital Endocasts: from Skulls to Brains*. 59–69. Eds: E. Bruner, N. Ogiwara & H. C. Tanabe. Tokyo: Springer.

Li 201 reference added Walsh & Knoll (2018)

Li. 203: How could you tell, in lateral view, that a structure is positioned more laterally?

Li 203 lateral view → caudal view

Li. 214: Well, this is problematic. I think you should at least provide a figure of the original endocast(that is, before such amendments are made).

Li 214 two referred models *Velociraptor* and *Zanabazar* are added. And the original 3D models are deposited in open data platform Dryad.

Li. 241: the labyrinth looks derived with respect to those of oviraptorosaurs, dromaeosaurids and some other troodontids (see e.g., Hanson et al., 2021).

Hanson M, Hoffman EA, Norell MA, Bhullar BS. The early origin of a birdlike inner ear and the evolution of dinosaurian movement and vocalization. *Science*, 372(6542): 601-609.

I beg to disagree about its resemblance with that of *Velociraptor*. In the latter, the RSCC is relatively much shorter, whereas in *Sinovenator* it is relatively much longer and arcuate. Thus, it extends dorsally much beyond the apex of the crus commune in *Sinovenator*, but not in *Velociraptor*. In this respect, the labyrinth of *Sinovenator* looks more bird-like. In lateral view, the vestibular system of the inner ear of *Sinovenator* does not look triangular at all to me. The lagena is remarkably robust, much more so than in *Velociraptor*.

Li. 241 the comment is very constructive. We rephrased the section by comparing the labyrinth with more taxa and added reference Hanson et al. (2021)

Li. 255: Could that be due to preservational issues?

Li. 255 the dorsoventrally compressed ampulla of posterior and lateral canal. We cannot totally eliminate the possibility from preservation, but the caudal part of the skull is mostly preserved intact, and the two ampullae are compressed in two different directions, thus their morphology is not likely to be caused by preservation.

Li. 259: “though this may be due...”: I don’t understand.

Li. 259 following Li. 255, we rephrased the entire sentence.

Li. 264: “with no evidence of twisting...”: I disagree, it is apparent that the lagena not only curves medially but also caudally.

Li. 264 labeled the lagena and other structures (utricle, saccule).

Li. 265: I disagree. See above.

Li. 265 as Li. 264

Li. 266: "Laterally...". Please, rephrase.

Li. 266 Laterally → Caudally

Li. 314: You can cite Walsh & Knoll (2018) here as well.

Li. 314 reference added Walsh & Knoll (2018)

Li. 317: tract, not tact.

Li. 317 tact → tract

Fig.: please provide dorsal and ventral views of the endocast. Also label each cranial nerve clearly.

For figures, we added ventral and dorsal view of the cranial endocast and rearrange the figure orders. The cranial nerves are labelled.

Reviewer #3 (Remarks to the Author):

Xu et al. describe the endocast of *Sinovenator* based on an exceptionally well-preserved skull belonging to this taxon. The authors interpret the similarity between the endocast of *Sinovenator* and *Archaeopteryx* as the result of convergence among paravian dinosaurs. Overall, I enjoyed the manuscript, which is well written and executed. I have only few major comments regarding the main interpretation of the findings, additional analyses, and sharing of the data.

Major comments:

- General interpretation of brain evolution: can the authors explain why this should be convergence and not being representative of the plesiomorphic condition for paravians? We don't currently have any endocast regarding small-bodied or early diverging dromaeosaurs (not published at least), exception made for *Bambiraptor* which is considered a juvenile. Maybe it is only me, but I could perfectly picture something like the endocast of *Sinovenator* or *Archaeopteryx* as an early paravian condition, rather than convergence among relatively closely related avialan and non-avian dinosaur. The authors kind of suggest this in the abstract, but I would love to hear more about the evolutionary scenario that they depict based on their novel results. I would also suggest the authors to run an evolutionary rate analyses using the PCoordinates and the assembled informal supertree to see if indeed this is a matter of convergence or not based on current data.

- Geometric morphometrics: I believe that the initial part of the paragraph in the results can be moved to the materials and methods. On the other hand, a sentence giving an idea of the number of taxa and their phylogenetic attribution might be useful here, considering that, without checking the figures, it is currently impossible to grasp an overview of the comparative dataset and sample size. Additionally, I find the overall description of what goes closer to what in the morphospace quite preliminary for the overall goal of the study (demonstrating convergence/morphological similarity between the endocast of early avialans and non-avian dinosaurs). I would require the authors to run a cluster analysis to actually evaluate if *Archaeopteryx*, *Sinovenator*, and the young troodontid fall within one single cluster or not.

- Data policy: Communication biology embraces open access. Yet there is no statement regarding how the CT scans and endocast volume will be treated after publication. I would kindly ask the authors to make at least the .stl files of the skull and endocast (including semicircular canal) freely available upon publication of the manuscript.

- Figure 4A is impossible to read. Please, make sure to improve the resolution of the figure before publication

Figure 4A is modified with enlarged text for better reading.

- I believe that important references are still missing from the manuscript, especially for comparative purposes. I would ask the authors to include the following, that I am sure they will find useful to expand

their discussion and comparative framework

References:

- 1) Alonso, P. D., Milner, A. C., Ketcham, R. A., Cookson, M. J., & Rowe, T. B. (2004). The avian nature of the brain and inner ear of Archaeopteryx. *Nature*, 430(7000), 666-669.
- 2) Beyrand, V., Voeten, D. F., Bureš, S., Fernandez, V., Janáček, J., Jirák, D., ... & Tafforeau, P. (2019). Multiphase progenetic development shaped the brain of flying archosaurs. *Scientific Reports*, 9(1), 10807.
- 3) Hanson, M., Hoffman, E. A., Norell, M. A., & Bhullar, B. A. S. (2021). The early origin of a birdlike inner ear and the evolution of dinosaurian movement and vocalization. *Science*, 372(6542), 601-609.
- 4) Fabbri, M., Mongiardino Koch, N., Pritchard, A. C., Hanson, M., Hoffman, E., Bever, G. S., ... & Bhullar, B. A. S. (2017). The skull roof tracks the brain during the evolution and development of reptiles including birds. *Nature ecology & evolution*, 1(10), 1543-1550.
- 5) Ksepka, D. T., Balanoff, A. M., Smith, N. A., Bever, G. S., Bhullar, B. A. S., Bourdon, E., ... & Smaers, J. B. (2020). Tempo and pattern of avian brain size evolution. *Current Biology*, 30(11), 2026-2036.

We added missing references and other references suggested by reviewer 3 and others, and also added more discussion based on these studies.

Minor comments:

- Line 93: the authors should make clear if these features were the original list recovered by Xu et al. (2002) or not. It is not clear at the moment

Line 93, we confirmed those features are from Xu et al. (2002) by rephrasing the sentence.

- The last paragraph of the introduction would be better suited as the second to last: the last paragraph sounds as a recapitulation of the state of the art regarding the cranial morphology of the investigated taxon. For this reason, I believe that the paragraph starting with “In this study, we...” should be placed at the end of the introduction

Last paragraph of introduction, rather than switching the order of the last two paragraph, we placed the sentence “In this study, we report” in the end of introduction and adjust the order of some sentences. The basic idea is following temporal order to introduce progress on *Sinovenator*/troodontid studies.

- Line 143: what is the correct shape of the foramen magnum according to the authors’ view and experience with these fossils? A clear indication regarding this matter is needed for future reference

Line 143 the shape of the foramen magnum in *Sinovenator* is more likely to be dorsoventrally taller than wide as suggested by Xu et al. (2002) and Yin et al. (2018). The sentence is revised.

Reviewers' comments:

Reviewer #1 (Remarks to the Author):

The ms by Yu and colleagues is an improved revision of a previously submitted ms on the endocranial anatomy of a specimen of *Sinoovenator*. The fossil material is amazing, the new interpretations are interesting and both the description and analyses are well performed (the ms is clearly written, only minor typos). Two comments:

1) I still think that it would be worth stressing in the introduction (perhaps in the last paragraph) that the authors perform a detailed description of the brain's endocast; namely, ten paragraphs of description might not only be to "report" on this new data, and it clearly overrides the morphometrics part.

2) I understand why, according to Watanabe et al., (2021), the authors explain the observed shape changes of the brain as "mosaic trends". However, the interpretation of such trends is discussed in terms of shape differences (e.g., which I guess it is what is meant in L347 by "dorsoventrally flexed brain architecture acquired by crow birds"), but the cited papers that justify such interpretation are those of Iwaniuk and Fong. These studies interpret the mosaic evolution of the brain only in terms of the relative evolutionary independence of brain parts' relative size increase (which is also called "concerted evolution" by neurobiologists such as Striedter 2004-Principles of Brain Evolution). However, relative size increase of parts might be independent of the topology (shape) of the brain, as a whole. There are myriad papers in mammals (some in birds) clearly explaining that brain part sizes, relative brain part sizes, and whole brain shapes, are different things, and indeed, that they might not be correlated. Literature in mammals is vast, especially in Paleoanthropology; see e.g., Bastir et al., 2011 Nat Comms; Brunner et al., 2015 JAnat and Brunner 2017 Digital Endocasts; in birds, see e.g., Marugán-Lobón et al. 2022 JAnat and Marugán-Lobón et al., 2018. Accordingly, I would suggest simply adding a sentence or two that clarifies whether the interpretation of such trends in dinosaurs refer to a) only the topological (or shape) evolutionary changes of the brain/endocast, b) to the relative size increase of brain parts across phylogeny, or c) both.

Minor details:

L63 Endocasts_are

L123 and_ (

L238 Two "at least"

L258 basal avialans

L272 thickest "."

L279 2021)..

L308 Strong evolutionary allometric signal? (1) R2 is extremely low (and, 0.0093 may not be that significant, depending on the replicas), and (2) The Shape-Space (PCA) actually shows that allometry is negligible, as discussed in the text (and surprisingly—the trends go in completely opposite directions).

L324 Analysed with various bird-line dinosaurs..."Compared"?

L337 Paulina is OK, the rest of citations says "Pauline". It was hard to follow the literature because in-text citations are in MLA but list order is in Chicago style (numbers).

L355 Comments on "mosaic trends", see above.

L363 As far as I understood, regression lines weren't depicted... only the Size-Shape Space is shown with ad hoc fitted lines (only for explanatory purposes). Please clarify.

L365 regressions

L368 Troodontids

L418 Given that the authors chose to use of Size-Shape space, I'd suggest briefly explaining how this is performed and what it means. Bookstein and Mitteroecker changed the terminology for the PCs in their paper, but it would be worth explaining to the reader why PC1 turns into the "Common Allometric Component" after the introduction of CS in the computation, and why this explains how PC2 thus becomes "Residual Shape". It would also clarify the difference with the regression that was performed.

Figure 5: I think it's Minimum Spanning tree. I'd suggest switching the score's signs in PC1Residual shape so that the graphic is congruent with that of Fig. 5A (left), as they are nearly identical. And perhaps make them the same size to emphasize this similarity (i.e., that PC1 is not allometric).

I've also read through the second reviewer's comments to remember what he said, and I think that his major concerns have been solved. Actually, they were largely similar to mine (i.e., The mere fact that the brain of Sinovenator closely resembles that of Archaeopteryx is not sufficient to sustain a claim of convergent evolution. I don't understand very well either the few lines of discussion about the "mosaic pattern"), and much of what he commented as improvements seem to have been revised by the authors.

Reviewer #3 (Remarks to the Author):

I find the new version of the manuscript by Yu et al. improved in comparison to the first draft, but I am kind of disappointed to see that my suggestions for the analyses were not taken into consideration and the rebuttal letter shows no answer to my questions. I would expect at least a justification of why my concerns were not taken into account. Additionally, not all the suggested references that were missing are present in the manuscript: I wonder why.

Unfortunately, I can not accept the current version of the manuscript as it is until my concerns will be addressed.

Reviewer #1 (Remarks to the Author):

The ms by Yu and colleagues is an improved revision of a previously submitted ms on the endocranial anatomy of a specimen of *Sinovenator*. The fossil material is amazing, the new interpretations are interesting and both the description and analyses are well performed (the ms is clearly written, only minor typos). Two comments:

1) I still think that it would be worth stressing in the introduction (perhaps in the last paragraph) that the authors perform a detailed description of the brain's endocast; namely, ten paragraphs of description might not only be to "report" on this new data, and it clearly overrides the morphometrics part.

We add one line in the last introduction paragraph that states this study includes qualitative anatomical description.

2) I understand why, according to Watanabe et al., (2021), the authors explain the observed shape changes of the brain as "mosaic trends". However, the interpretation of such trends is discussed in terms of shape differences (e.g., which I guess it is what is meant in L347 by "dorsoventrally flexed brain architecture acquired by crow birds"), but the cited papers that justify such interpretation are those of Iwaniuk and Fong. These studies interpret the mosaic evolution of the brain only in terms of the relative evolutionary independence of brain parts 'relative size increase (which is also called "concerted evolution" by neurobiologists such as Striedter 2004-Principles of Brain Evolution). However, relative size increase of parts might be independent of the topology (shape) of the brain, as a whole. There are myriad papers in mammals (some in birds) clearly explaining that brain part sizes, relative brain part sizes, and whole brain shapes, are different things, and indeed, that they might not be correlated. Literature in mammals is vast, especially in Paleoanthropology; see e.g., Bastir et al., 2011 Nat Comms; Brunner et al., 2015 JAnat and Brunner 2017 Digital Endocasts; in birds, see e.g., Marugán-Lobón et al. 2022 JAnat and Marugán-Lobón et al., 2018. Accordingly, I would suggest simply adding a sentence or two that clarifies whether the interpretation of such trends in dinosaurs refer to a) only the topological (or shape) evolutionary changes of the brain/endocast, b) to the relative size increase of brain parts across phylogeny, or c) both.

To address the reviewer's comment, shape differences encompasses changes in proportional sizes of parts of endocasts and relative positions. However, we are unable to confirm the degree of mosaic or concerted trends in shape data without performing a modularity & integration analysis, which is not feasible with so few points representing each a priori partitions (e.g., "cerebrum", "optic lobe", "cerebellum", "medulla" in Watanabe et al. 2021). For this reason, we have limited our statements about mosaic evolution to qualitatively describe the co-occurrence of the plesiomorphic configuration of the olfactory apparatus and the brain dorsoventral flexion.

Minor details:

L63 Endocasts_are
Space inserted

L123 and_ (
“and ” deleted

L238 Two “at least”
2nd “at least” deleted

L258 basal avialans
basal avianlan → basal avialans
also L261 canal in → canal in, space deleted

L272 thickest “.”
“.” deleted

L279 2021)..
“.” Deleted

L308 Strong evolutionary allometric signal? (1) R² is extremely low (and, 0.0093 may not be that significant, depending on the replicas), and (2) The Shape-Space (PCA) actually shows that allometry is negligible, as discussed in the text (and surprisingly—the trends go in completely opposite directions).

We have removed the word “strong” when describing the degree of allometric signal in endocast shape data. R-squared of 0.163 is consistent with typical value seen with geometric morphometric data.

L324 Analysed with various bird-line dinosaurs...”Compared”?
Analysed → compared

L337 Paulina is OK, the rest of citations says “Pauline”. It was hard to follow the literature because in-text citations are in MLA but list order is in Chicago style (numbers).
All Pauline corrected to Paulina, And the order of reference follows MLA.

L355 Comments on “mosaic trends”, see above.
We followed the suggestion and added one sentence before this.

L363 As far as I understood, regression lines weren’t depicted... only the Size-Shape Space is shown with ad hoc fitted lines (only for explanatory purposes). Please clarify.

L365 regressions

regression → regressions

L368 Troodontids

troodontid → troodontids

L418 Given that the authors chose to use of Size-Shape space, I'd suggest briefly explaining how this is performed and what it means. Bookstein and Mitteroecker changed the terminology for the PCs in their paper, but it would be worth explaining to the reader why PC1 turns into the "Common Allometric Component" after the introduction of CS in the computation, and why this explains how PC2 thus becomes "Residual Shape". It would also clarify the difference with the regression that was performed.

We add more explanation regarding Figure 5B in L320. Essentially, the fact that the original Fig. 5A and 5B (now Fig. 5A and 5C) resemble one another means that PC1 is very highly correlated with size and because PC1 of residuals is essentially PC2 (they are both orthogonal dimension to the PC1/CAC), that is why Fig. 5A and 5B look nearly identical.

Figure 5: I think it's Minimum Spanning tree. I'd suggest switching the score's signs in PC1 Residual shape so that the graphic is congruent with that of Fig. 5A (left), as they are nearly identical. And perhaps make them the same size to emphasize this similarity (i.e., that PC1 is not allometric).

We thank the reviewer for an insightful comment. We have swapped the vertical axis of Fig. 5B (now C) to match the configuration of Fig. 5A.

I've also read through the second reviewer's comments to remember what he said, and I think that his major concerns have been solved. Actually, they were largely similar to mine (i.e., The mere fact that the brain of Sinovenator closely resembles that of Archaeopteryx is not sufficient to sustain a claim of convergent evolution. I don't understand very well either the few lines of discussion about the "mosaic pattern"), and much of what he commented as improvements seem to have been revised by the authors.

We agree with the point that the similarities in endocranial shape does not imply convergence. We have removed statements referring to the shape similarity as being due to convergent evolution.

Reviewer #3 (Remarks to the Author):

I find the new version of the manuscript by Yu et al. improved in comparison to the first draft, but I am kind of disappointed to see that my suggestions for the analyses were not taken into consideration and the rebuttal letter shows no answer to my questions. I would expect at least a justification of why my concerns were not taken into account. Additionally, not all the suggested references that were missing are present in the manuscript: I wonder why.

Unfortunately, I can not accept the current version of the manuscript as it is until my concerns will be addressed.

We apologize for some misunderstandings in the communication process between our authors, and between authors and editors. And then apologies for overlooking some of previous revisions during communications, and comments we have collated reviewer 3's comments from the last review and responded below.

Comments from Reviewer #3 in 1st revision and reply:

Xu et al. describe the endocast of *Sinovenator* based on an exceptionally well-preserved skull belonging to this taxon. The authors interpret the similarity between the endocast of *Sinovenator* and *Archaeopteryx* as the result of convergence among paravian dinosaurs. Overall, I enjoyed the manuscript, which is well written and executed. I have only few major comments regarding the main interpretation of the findings, additional analyses, and sharing of the data.

Major comments:

1. General interpretation of brain evolution: can the authors explain why this should be convergence and not being representative of the plesiomorphic condition for paravians? We don't currently have any endocast regarding small-bodied or early diverging dromaeosaurs (not published at least), exception made for *Bambiraptor* which is considered a juvenile. Maybe it is only me, but I could perfectly picture something like the endocast of *Sinovenator* or *Archaeopteryx* as an early paravian condition, rather than convergence among relatively closely related avialan and non-avian dinosaur. The authors kind of suggest this in the abstract, but I would love to hear more about the evolutionary scenario that they depict based on their novel results. I would also suggest the authors to run an evolutionary rate analyses using the PCoordinates and the assembled informal supertree to see if indeed this is a matter of convergence or not based on current data.

The reviewer is correct that the interpretation of the similar endocast shape as due to convergent/independent evolution is not well supported. We have removed statements interpreting this similarity as convergence and have described it, instead, as intermediary morphotype between non-avian dinosaurs and crown birds. In addition, conducting a rate analysis with such limited taxonomic sampling (due to limited number of three-dimensionally intact braincase of paravian dinosaurs that also have been CT scanned) is unlikely to generate biologically reliable results. For these reasons, we have decided to not perform an evolutionary rates analysis but rather a UPGMA clustering based on pairwise Euclidean distances (Fig. 5B) to confirm the similarity between *Sinovenator* and *Archaeopteryx* endocast.

To validate our claim, we performed evolutionary rate analysis but the result is not included in the manuscript. We estimated the evolutionary rate on the basis of a variable-rate Brownian Motion (BM) evolution model. BayesTraits (<http://www.evolution.edg.ac.uk/>) variables rates analyses were performed to estimate the rate of endocast shape evolution across our sampled taxa. The phylogenetic tree topology is on the basis of Gold and Watanabe (2018) and Watanabe et al. (2021) with *Sinovenator* added at the position of the most basal troodontid (Xu et al. 2002). Variable-rates models were performed with first 13 PC axes scores that account for more than

95% of total variation. Variable-rates BM evolution model with 100,000,000 iterations and a burn-in of 12,500,000 were selected to make sure the Markov Chain has reached convergence. The results show generally low rate across most non-avian theropod dinosaurs, although the lineages of *Sinovenator* and *Archaeopteryx* show slightly higher rates, the relatively small sample size and phylogenetic gaps between taxa make the results less meaningful.

2. Geometric morphometrics: I believe that the initial part of the paragraph in the results can be moved to the materials and methods. On the other hand, a sentence giving an idea of the number of taxa and their phylogenetic attribution might be useful here, considering that, without checking the figures, it is currently impossible to grasp an overview of the comparative dataset and sample size. Additionally, I find the overall description of what goes closer to what in the morphospace quite preliminary for the overall goal of the study (demonstrating convergence/morphological similarity between the endocast of early avialans and non-avian dinosaurs). I would require the authors to run a cluster analysis to actually

evaluate if Archaeopteryx, Sinovenator, and the young troodontid fall within one single cluster or not.

We have indicated the sample size for non-avian theropod dinosaurs and modern birds in the Introduction section to clarify the sample size for each taxonomic group. We also performed a clustering analysis (UPGMA) based on distance matrix of the full shape data (Fig. 5C).

3. Data policy: Communication biology embraces open access. Yet there is no statement regarding how the CT scans and endocast volume will be treated after publication. I would kindly ask the authors to make at least the .stl files of the skull and endocast(including semicircular canal) freely available upon publication of the manuscript.

[AW comment to Congyu/Xu/Zichuan: could the mesh file of the endocasts be provided as SI or uploaded to data repository like Dryad or MorphoSource?]

We have uploaded the 3D model into Dryad in last revision at DOI: 10.5061/dryad.41ns1rnk6. We have clarified this in the supplementary materials section.

4. Figure 4A is impossible to read. Please, make sure to improve the resolution of the figure before publication

We have uploaded Figs in high resolution, those embedded in .docx files are not for final publication.

5. I believe that important references are still missing from the manuscript, especially for comparative purposes. I would ask the authors to include the following, that I am sure they will find useful to expand their discussion and comparative framework

References:

- 1) Alonso, P. D., Milner, A. C., Ketcham, R. A., Cookson, M. J., & Rowe, T. B. (2004). The avian nature of the brain and inner ear of Archaeopteryx. *Nature*, 430(7000), 666-669.
- 2) Beyrand, V., Voeten, D. F., Bureš, S., Fernandez, V., Janáček, J., Jirák, D., ... & Tafforeau, P. (2019). Multiphase progenetic development shaped the brain of flying archosaurs. *Scientific Reports*, 9(1), 10807.
- 3) Hanson, M., Hoffman, E. A., Norell, M. A., & Bhullar, B. A. S. (2021). The early origin of a birdlike inner ear and the evolution of dinosaurian movement and vocalization. *Science*, 372(6542), 601-609.
- 4) Fabbri, M., Mongiardino Koch, N., Pritchard, A. C., Hanson, M., Hoffman, E., Bever, G. S., ... & Bhullar, B. A. S. (2017). The skull roof tracks the brain during the evolution and development of reptiles including birds. *Nature ecology & evolution*, 1(10), 1543-1550.
- 5) Ksepka, D. T., Balanoff, A. M., Smith, N. A., Bever, G. S., Bhullar, B. A. S., Bourdon, E., ... & Smaers, J. B. (2020). Tempo and pattern of avian brain size evolution. *Current Biology*, 30(11), 2026-2036.

We added missing references and other references suggested by reviewer 3 and others, and also added more discussion based on these studies.

Minor comments:

6. Line 93: the authors should make clear if these features were the original list recovered by Xu et al. (2002) or not. It is not clear at the moment

Line 93, we confirmed those features are from Xu et al. (2002) by rephrasing the sentence.

7. The last paragraph of the introduction would be better suited as the second to last: the last paragraph sounds as a recapitulation of the state of the art regarding the cranial morphology of the investigated taxon. For this reason, I believe that the paragraph starting with “In this study, we...” should be placed at the end of the introduction

Last paragraph of introduction, rather than switching the order of the last two paragraph, we placed the sentence “In this study, we report” in the end of introduction and adjust the order of some sentences. The basic idea is following temporal order to introduce progress on *Sinovenator*/troodontid studies.

8. Line 143: what is the correct shape of the foramen magnum according to the authors’ view and experience with these fossils? A clear indication regarding this matter is needed for future reference

Line 143 the shape of the foramen magnum in *Sinovenator* is more likely to be dorsoventrally taller than wide as suggested by Xu et al. (2002) and Yin et al. (2018). The sentence is revised.

REVIEWERS' COMMENTS:

Reviewer #1 (Remarks to the Author):

The authors have made important changes in the ms and I am glad that they agreed with most of my suggestions. It is an interesting paper.

Reviewer #3 (Remarks to the Author):

The authors addressed all the concerns and comments, changing the manuscript accordingly. I strongly suggest publication of this study in its current form